# Visual Function and Neuropsychological Profiling of Idiopathic Infantile Nystagmus

**DOI:** 10.3390/brainsci13091348

**Published:** 2023-09-20

**Authors:** Federica Morelli, Guido Catalano, Ilaria Scognamillo, Nicolò Balzarotti, Antonella Luparia, Lucrezia Olivier, Chiara Bertone, Monica Gori, Sabrina Signorini

**Affiliations:** 1Department of Brain and Behavioral Sciences, University of Pavia, Via Agostino Bassi 21, 27100 Pavia, Italy; federica.morelli02@universitadipavia.it (F.M.); ilaria.scognamillo01@universitadipavia.it (I.S.); 2Developmental Neuro-Ophthalmology Unit, IRCCS Mondino Foundation, Via Mondino 2, 27100 Pavia, Italy; antonella.luparia@mondino.it (A.L.); lucrezia.olivier@mondino.it (L.O.); sabrina.signorini@mondino.it (S.S.); 3Unit for Visually Impaired People, Istituto Italiano di Tecnologia, Via Enrico Melen 82, 16100 Genova, Italy; nicolo.balzarotti@iit.it (N.B.); monica.gori@iit.it (M.G.); 4Department of Surgical and Clinical, Diagnostic and Pediatric Sciences, Section of Ophthalmology, IRCCS Fondazione Policlinico San Matteo, University of Pavia, 27100 Pavia, Italy; c.bertone@smatteo.pv.it

**Keywords:** idiopathic infantile nystagmus, vision, visual impairment, neurodevelopment, visual–motor integration, WISC-IV, Beery-VMI, dorsal stream vulnerability

## Abstract

Though considered a benign condition, idiopathic infantile nystagmus (IIN) may be associated with decreased visual acuity and oculo-motor abnormalities, resulting in developmental delays and poor academic performance. Nevertheless, the specific visual function profile of IIN and its possible impact on neuropsychological development have been poorly investigated. To fill this gap, we retrospectively collected the clinical data of 60 children presenting with IIN over a 10-year period (43 male; mean age of 7 years, range of 2 months-17 years, 9 months). The majority of the subjects in our cohort presented with reduced visual acuity for far distances and normal visual acuity for near distances, associated with oculo-motor abnormalities. The overall scores of cognitive and visual–cognitive tests were in the normal range, but revealed peculiar cognitive and visual–cognitive profiles, defined by specific frailties in processing speed and visual–motor integration. The same neuropsychological profiles characterize many neurodevelopmental disorders and may express a transnosographic vulnerability of the dorsal stream. As the first study to explore the neuropsychologic competencies in children with IIN, our study unveils the presence of subclinical frailties that need to be addressed to sustain academic and social inclusion.

## 1. Introduction

Infantile nystagmus (IN) is defined as an involuntary rhythmic oscillation of the eyes, which occurs in the first 6 months of age. It was traditionally known as *congenital nystagmus*, though the term *infantile* is now preferred as nystagmus is not always present at birth [1,2]. Based on neuro-ophthalmological and electrophysiological findings, IN can be classified into three conditions [3,4,5]: (a) sensory nystagmus (i.e., associated with ocular conditions such as retinal diseases or congenital cataract), (b) neurological nystagmus (i.e., associated with neurological/neuroanatomical abnormalities such as periventricular leukomalacia, brain malformations, or neuro-developmental syndromes), and (c) idiopathic infantile nystagmus (IIN). In the absence of overt neurological or ocular conditions, IIN usually manifests in the first 6 months of age with irregular eye movements and is usually a diagnosis of exclusion [6]. A wide epidemiologic study conducted in England found a total prevalence of 1.9 per 10,000 in a population of unassociated IN patients (i.e., nystagmus in patients with negative ocular and electrodiagnostic tests), equal to 20% of nystagmus in childhood overall [7]. Accordingly, a recent Italian study on paediatric nystagmus reported a prevalence of 22% for IIN out of 132 children presenting with nystagmus [8]. IIN incidence in a retrospective, population-based study by Nash et al. was of 6.72 per 100,000 live births [9]. Clinically, idiopathic infantile nystagmus is usually conjugate, jerk, and horizontal [5]. Its aetiology is not completely known yet, though there is a relative consensus on its genetic origin. Both sporadic and inherited (autosomal dominant, recessive, X-linked) forms have been reported, with FRMD7, GPR143 and CASK considered causative genes [10,11]. Nevertheless, the underlying pathogenic mechanisms have not been clarified yet. The functional disruption of neuronal systems involved in ocular motility (fixation, pursuit, and saccades) has been proposed and is sustained by advanced imaging techniques [2,4,12]. Mild retinal developmental abnormalities have been recently reported in children with FRMD7 pathogenic mutations [13,14], supporting a sensory hypothesis for the IIN origin. Brodsky et al. proposed a unifying hypothesis: the slow development of foveation would alter the maturation of the cortical pursuit and fail to inactivate the accessory optic system, a primitive subcortical structure [15]. Though no overt ocular or neurological manifestations are present, IIN is often associated with reduced visual acuity because of the excessive motion of images on the retina, and the shifting of images away from the fovea [16]. The possible detrimental effect of congenital visual impairment on development has been extensively studied [17], and some authors have also investigated its anatomo-functional correlates. For example, Bathelt et al. demonstrated a reduction in neuroanatomical brain structures in the visual systems of children affected by congenital visual impairment. This finding would sustain the hypothesis that the organization of central visual structures might be influenced by the quantity or quality of sensory (visual) inputs during development [18]. Holding with these premises, one could wonder whether a condition such as IIN (i.e., non-degenerative and not associated with neurologic involvement) may have similar consequences due to inconstant foveation.

It is already known how IIN affects predominantly ocular motor abilities [19,20,21], while the literature has scarcely focused on the involvement of other visual functions, apart from some reports concerning, in particular, velocity discrimination [22], spatial bisection acuity [23], motion perception [24], and visual crowding [25,26]. Such dysfunctions may come along with a disruption in the development of cognitive competencies and a negative repercussion on functional vision (i.e., how the child functions in everyday life vision-related tasks, including academic abilities). Barot et al., for example, reported a reduced reading speed in children with IIN using non-adapted font sizes [27]. Nevertheless, to our knowledge, there are only a few studies that consider the overall neuropsychological profiles (including cognitive, visual–cognitive, and learning abilities) of children with IIN. Exploring the possible impact of visual functions on development could help (a) delineate a comprehensive profile of children with IIN and (b) provide insights for specific assessments and training to promote academic inclusion and success. Furthermore, the visual function and neuropsychological profiling of this population could pave the way for a better understanding of the underlying functional correlates of such a condition.

This paper aims to describe the clinical, visual, and neuropsychological characteristics of a cohort of children affected by IIN to detect peculiarities in their neuropsychological (cognitive and visual–cognitive) profile. As a secondary goal, it aims to evaluate whether basic visual functions (such as best-corrected visual acuity, fixation, smooth pursuit, and saccades) influence the development of neuropsychological skills in children with IIN.

## 2. Materials and Methods

### 2.1. Participants

We conducted a retrospective analysis on a cohort of paediatric patients referred to the developmental neuro-ophthalmology unit of a tertiary referral hospital for neurological conditions (IRCCS Mondino Foundation, Pavia, Italy) from 1 January 2007 to 31 May 2023. We included subjects < 18 years of age who had received a diagnosis of IIN. A diagnosis of exclusion was defined according to guidelines [1,4,5] and relied on familiar and medical history, a neurological and ophthalmologic examination (including a visual function evaluation), and diagnostic exams such as electroretinogram (ERG) and visual evoked potentials (VEP) according to the ISCEV protocols [28,29], and brain magnetic resonance imaging (MRI). All of these exams were negative for the tested patients. The neuro-ophthalmologic and neuropediatric exams during the follow-up confirmed the normal findings. Where available, genetic next-generation sequencing (NGS) examination data were collected for confirmation (see Appendix A for details). Optical coherence tomography (OCT) was not performed in our cohort. OCT is not frequently available nor easy to perform in paediatric contexts, being even less reliable when the fixation is impaired, and IIN has been diagnosed in the absence of such an examination [8,30].

We collected data concerning the clinical details and neuro-ophthalmological evaluations of 60 children affected by IIN and aged between 2 months and 17 years (see Table 1 for the demographic characteristics of the cohort). All the evaluations were based on each child’s age and performed for clinical purposes by a multidisciplinary team of professionals, including child neuropsychiatrists, ophthalmologists, developmental therapists, and neuropsychologists with expertise in the field.

### 2.2. Methods

The visual functions are summarized in Table 2. We collected and categorized the data according to the standardized clinical protocol presented by Signorini et al. that has already been used in research papers [31,32]. Nystagmus was clinically evaluated through a video recording and a multidisciplinary (both neuro-ophthalmological and neuro-paediatric) assessment.

The neuropsychological (cognitive, visual–cognitive, and learning abilities) assessment was based on the following tests, chosen based on the age of the subject. When necessary, a bookrest was used to guarantee a correct posture. The patient selection for the neuropsychological profile analysis followed the procedure reported in the following flow chart (Figure 1).

-The Wechsler preschool and primary scale of intelligence, third edition (WPPSI-III) [33], Wechsler preschool and primary scale of intelligence, fourth edition (WPPSI-IV) [34], Wechsler intelligence scale for children, fourth edition (WISC-IV) [35], and Wechsler adult intelligence scale, fourth edition (WAIS-IV) [36]: For the WISC-IV and WAIS-IV, we considered the following scores: (i) the verbal comprehension index (VCI); (ii) the perceptual reasoning index (PRI); (iii) the working memory index (WMI); (iv) the processing speed index (PSI); and (v) the intelligence quotient (IQ). For the WPPSI-III, we collected the following scores: (i) the verbal comprehension index (VCI); (ii) the performance index (PI); (iii) the processing speed index (PSI); and (iv) the total intelligence quotient (TIQ). For the WPPSI-IV, we collected the following scores: (i) the verbal comprehension index (VCI); (ii) the visual–spatial index (VSI) (iii); the fluid reasoning index (FRI) (iv) the working memory index (WMI); (v) the processing speed index (PSI); and (vi) the full-scale IQ (FSIQ).-The Beery developmental test of visual–motor integration (Beery-VMI) [37], composed of the subtests visual perception (VMI-V) and motor coordination (VMI-M): The results were collected in terms of the standard scores and categorized according to the percentiles as normal (>16°P), frail (5°–16°P), or deficient (<5°P). The VMI task evaluates the integration of visual perception and motor skills as the examinee imitates and copies a developmentally sequential series of geometric forms using a pencil and paper. The VMI-V task evaluates an individual’s visual abilities without the integration of fine motor skills. The VMI-M task evaluates fine motor skills when not specifically integrated with visual perceptual abilities [38] (see Figure 2).-The developmental test for visual perception (DTVP) [37]: The general visual perceptual (DTVP-GVP), non-motor visual perceptual (DTVP-NMVP), and visual–motor integration (DTVP-VMI) quotients were collected and categorized according to the percentiles as normal (>16°P), frail (5°–16°P), or deficient (<5°P). The DTVP-2 consists of eight subscales, four of which assess motor-free visual perceptual skills (also referred to as motor-reduced) and four of which assess visual–motor integration.

Learning abilities (reading, writing, and math) were evaluated using specific tests validated for the Italian population, such as DDE-2 (battery for dyslexia and developmental dysorthography) and MT-3 test [39,40]. The reading tests were adapted to the children’s visual profile in terms of font size and line spacing.

For a specific study of cognitive profiles, we selected a subgroup of patients who performed the same test (WISC-IV; *n* = 18). A visual–cognitive assessment was available for 27 patients who performed the same tests (Beery-VMI, DTVP).

### 2.3. Data Analysis and Statistics

Data were analysed using the free software R, version 4.3.0 [41]. For each test and visual function parameter, we report the number and percentage of observations for each possible answer. For learning abilities, we report the percentage of patients whose score was normal or deficient. To evaluate whether our sample had sufficient power to compute a statistical analysis on each dependent variable (i.e., cognitive, visual-cognitive, and learning), we computed the power analysis by calculating the sample size on the free software G*Power 3.1 [42], based on similar papers on different populations [43,44,45]: − effect size, dz: 1.0; tails = two; − α err. prob. = 0.05; and − power (1-β err. prob.) = 0.95. The calculated sample size was 16. Therefore, we excluded from the analysis comparisons in which the sample size was <16. We compared the means between the subscales of the neuropsychological tests with a sample size ≥ 16 (Beery-VMI, WISC-IV, and DTVP). We used the parametric *t*-test after having verified that the distribution of each variable was not significantly different from the normal distribution (*p*-values of the Shapiro–Wilk test > 0.1 and visual inspection of qqplots). The *p*-value was corrected using the Bonferroni method for multiple comparisons. For those tests, we also report descriptive statistics, including: the mean, median, range, and standard deviation. We performed linear regression models between the visual function parameters and the neuropsychological tests. No significant differences were found after the Bonferroni correction; therefore, the results are not reported.

## 3. Results

### 3.1. Visual Functions and Cognitive Profiles

First, we described the basic visual functions and cognitive profiles of the cohort (see Table 2 and Appendix A).
brainsci-13-01348-t002_Table 2Table 2Visual function characteristics of the cohort scored according to visual function score [31]. Best corrected visual acuity (BCVA) was assessed using line tests (Snellen optotype or LEA symbols test [46]. Grating acuity was assessed using Teller acuity visual charts [47] based on the age.Visual FunctionParameterNumber of Subjects%Nys waveformJerk4473.3Pendular610.0Mixed1016.7Nys directionHorizontal5591.7Vertical23.3Mixed (horizontal and vertical)35.0Visual acuity (for far distance)Not performed for age (grating acuity)1423.3Normal1016.7Mild low vision (0.5–0.7 logMAR)610Moderate low vision (0.7–1 logMAR)35Severe low vision (>1 logMAR)2745Visual acuity (for near distance) Not performed for age (grating acuity)1423.3Normal3761.6Mild low vision (0.5–0.7 logMAR)711.5Moderate low vision (0.7–1 logMAR)23.2Grating acuity Not necessary for age4676.7Normal11.7Reduced (standard distance for age)23.33Reduced (only testable for lower distances compared to standard age)1118.3Head tiltAbsent813.3Inconstant/variable711.7Mild head tilt1626.7Severe head tilt2948.3Visual axis alignmentNormal4778.3Mild misalignment with alternating fixation1220Paralytic misalignment11.7FixationStable, durable, binocular; no difference between near and distant1830.0Durable, but not binocular and/or alternating and/or durable, but slightly different from near and distant1830.0Instable/slightly discontinuous and/or different from near and distant, but sufficiently durable2236.7Fluctuating/eccentric23.3SaccadesFluid; complete; normal latency, conjugacy, and precision; no evident hypo- or hypermetria711.7Fluid, incomplete, and/or asymmetric and/or not binocular813.3Slight alteration (hypo- or hypermetria, fluidity, latency)2135.0Moderate alteration (hypo- or hypermetria, fluidity, latency)2033.3Severe alteration (hypo- or hypermetria, fluidity, latency)11.7Sporadic/difficult to elicit (conditioned by attention)11.7Not testable due to lack of cooperation or too severe of a clinical picture23.3Smooth pursuitDurable, complete, and binocular46.7Durable, but incomplete/asymmetric/non binocular58.3Slightly discontinuous in all or great parts of directions2846.7Discontinuous/jerky/augmented latency2135.0Inconstant/eccentric/fragmented23.3StereopsisNot testable for age (<6mo)23.3Present1525Partial 915Absent1423.3Not testable due to lack of cooperation or too severe of a clinical picture711.7Missing data1321.7

The results of the cognitive tests for the 27 patients who performed an IQ assessment are reported in Appendix A. A total of 24/27 IQ results were interpretable. A total of 23/24 IQ results were normal (total IQ > 85).

For the statistical analysis, we considered neuropsychological variables with a total sample greater than 16 subjects (see data analysis and statistics section). Consequently, we excluded from the analysis the Wechsler preschool and primary scale of intelligence (WPPSI-III/WPPSI-IV) and the Wechsler adult intelligence scale (WAIS-IV). The 18 patients (12 M, 6 F) who performed the WISC-IV were included in the cognitive profile analysis. Three total IQ results were not interpretable due to excessive sub-indices discrepancy. Overall, the patients performed better in the VCI and PRI than the PSI and WMI. Five patients had a PSI index < 85 (range: 62–115) and one patient had a total IQ < 85 (IQ = 76) (see Table 3 and Appendix A). In order to draw a cognitive profile of our cohort, we performed *t*-tests between each index mean. The PSI results were significantly lower than the VCI (*p* < 0.03) and PRI (*p* < 0.036) results (see Figure 3).

### 3.2. Visual–Cognitive Profiles

Concerning the neuropsychological tests assessing visual cognition, most of the cohort had normal performances in all subscales (see Table 4). The Beery-VMI showed lower performances in the subtests regarding motor (VMI-M; *p* < 3.3 × 10^−5^) and visual–motor (VMI; *p* < 9.5 × 10^−5^) performances than the pure perceptual subtests (VMI-V) (see Figure 4). The DTVP did not report such significative differences between the motor and non-motor subtests (*p* > 0.07).

### 3.3. Learning Abilities

A total of 20 patients (aged > 6 years according to guidelines) performed learning ability tests. A total of 65% presented with at least one specific learning disability (reading, writing, or math). Table 5 reports the distribution of learning ability deficits across our cohort.

None of the visual functions significantly influenced the neuropsychological performance variables of our cohort (*p* > 0.005).

## 4. Discussion

Idiopathic infantile nystagmus is the most frequent form of congenital nystagmus and is considered a benign, isolated visual condition. Nevertheless, it may be associated with reduced visual acuity and an impact on everyday life’s activities [16]. The reports of the consequences of IIN on neuropsychological or academic abilities are anecdotal, are inconclusive, and mainly concern reading [27,48]. To contribute to filling this gap, this work aimed to (a) describe the neuropsychological characteristics (including cognitive, visual–cognitive, and learning abilities) of a cohort of children affected by IIN and (b) explore the possible associations between visual function and functional vision (evaluated in terms of visual–cognitive and learning abilities) [49]. Investigating functional vision competencies in IIN would help define specific developmental profiles (with their strengths and weaknesses) and provide insights for an assessment and intervention. In fact, to date, there are no specific healthcare recommendations, except for an ophthalmologic follow-up and the use of an adequate refractive correction [16].

IIN is a diagnosis of exclusion, requiring a full phenotyping that includes clinical (ophthalmologic and neurologic) and electrophysiological examinations as well as brain MRI and OCT when necessary [3,4,5,30]. Nevertheless, exams such as ERG and OCT are not routinely available, as they require expertise in their administration and a level of cooperation that may be difficult to obtain in paediatric settings, especially when a child has an impaired fixation [8,50]. For this work, we ruled out ophthalmological and neurological conditions associated with infantile nystagmus based on a clinical follow-up, an ERG and VEP (which excluded overt pre- and retro-geniculate conditions and only showed aspecific prolongation previously reported in IIN [30,51]), a brain 3T-MRI (when necessary), and genetic testing (see Appendix A for details on the diagnostic exams). Holding with these premises, we considered our cohort as presumably affected by IIN until proven otherwise.

### 4.1. Visual Function Profiling

As expected, nystagmus was predominantly jerk, unidirectional (horizontal) [52], and associated with moderate to severe low vision for far distances [16]. The visual performance physiologically depends on three factors: (i) the retinal image slip velocities, (ii) foveation, and (iii) the state of the health of the eye and visual pathways [5]. Considering that IIN predominantly affects foveation, our results on the reduced BCVA and grating acuity are consistent with the literature [6]. Furthermore, the majority of the children in our cohort presented with a head tilt, a compensatory head posture allowing the use of a null position to improve foveation and target recognition [53]. A mild to moderate impairment of oculomotor abilities (fixation, saccades, and pursuit) was reported for most of our cohort and is consistent with the literature [4]. A thorough assessment of oculomotor abilities should be recommended due to the negative influence of oculomotor impairment on vision (see, for example, an eye-tracking study by Pel et al. [54]) and on the cognitive and academic performances [55,56].

### 4.2. Cognitive and Visual–Cognitive Profiling

Our study confirms that IIN children show a normal neuropsychomotor development and IQ, consistent with the absence of CNS involvement [4,52,57]. Nevertheless, clinical experience and the known possible effects of visual (perceptual and oculomotor) impairment on different competencies [4,27,52] suggested to us to deepen the neuropsychological profiles in homogeneous subgroups in order to identify possible specific patterns. Similarly to the total IQ, the results of the WISC-IV indices (verbal comprehension index, VCI; perceptual reasoning index, PRI; processing speed index, PSI; and working memory index, WMI) were in the normal range. Nevertheless, a comparison between the indices showed that the mean PSI was significantly lower than both the mean PRI and the VCI (*p* < 0.036 and *p* < 0.003, respectively). The PRI is related to fluid intelligence (Gf), defined as the ability to generate, transform, and manipulate different types of novel information in real time, while the VCI concerns crystalized intelligence (Gc), i.e., the ability to deduce secondary relational abstractions by applying previously learned primary relational abstractions [58,59]. The PSI is the result of the scores of subtests such as the coding (a motorically demanding task in which the patient has to write a line of symbols, each one corresponding to a number) and symbol search (in which the child is asked to identify a target symbol among other symbols) subtests [60]. The performances in both tasks may be influenced by visual crowding (i.e., the disruption of the identification and recognition process of single targets or a group of objects) [61,62] due to ineffective foveation and unsuccessful visual search due to oculomotor abnormalities. In fact, in both subtests, saccade integrity and fluency (which were generally altered in our cohort) are required. The literature is scarce about the relationship between oculomotor impairment and a low processing speed. For example, a recent study found a low PSI in children with schizophrenia and oculomotor impairment [63]. Interestingly, lower PSI scores have been reported in many neurodevelopmental disorders, such as learning disabilities, ADHD, or high-functioning ASD [62,64,65,66]. A paper by Mayes and Calhoun comparing a clinical population of 886 children with normal intelligence and 149 typically developing (non-clinical) children demonstrated better performances for control children over ADHD and ASD patients in all the explored areas (learning, attention, grapho-motor, and processing speed) [44]. These findings suggest that the PSI could represent a sensitive (though not specific) neuropsychological indicator of neurodevelopmental vulnerability.

Concerning visual–cognitive skills, visual–motor integration is the ability to integrate visual (perceptual) information and movement. It represents a composite brain function requiring visual attention, detection and identification, anticipatory judgment, motor planning, and motor execution. In physiological conditions, the perceptual visual system conveys information from the surrounding environment to cerebral associative areas (ventral and dorsal streams), where it is integrated with cognitive and motor inputs to enable appropriate actions [66]. The correct development of visual cognitive competencies thus requires the integrity of all the involved systems (namely the peripheral visual system, the associative and cortical areas, and the motor system) [67]. When these complex interactions are impaired, a child may present with specific (visual) cognitive frailties and struggle in academic tasks, especially in reading and math [68,69], even with a normal intelligence. While visual–motor integration deficits are well known in CNS disorders, such as CVI [56,70], to our knowledge, no studies have examined this competence in IIN, which represents a paradigmatic condition free from overt ocular or cerebral involvement, but associated with an altered perception and ocular motility. In our study, we explored visual–motor integration (VMI) using a standardized test, the Beery-VMI, which has been extensively used to assess children and adults with various developmental disorders and acquired brain injuries [71,72] (see Figure 2 and Figure 4). For the first time, we report a peculiar VMI profile in children with IIN. Indeed, the children in our cohort performed significantly worse in motor and visual–motor integration tasks compared to visual-only tasks (see Figure 2 and Figure 4). The DTVP test results are slightly above the significancy level, but show the same tendency. Such a result would suggest that children with IIN may experience difficulties during VMI tasks that primarily require an involvement of associative areas and the efferent (motor and oculomotor) systems. Moreover, our results suggest that perceptual aspects may play a minor role. Indeed, the BCVA for near distance in our cohort was in the normal–near-normal range in the majority of the subjects, and visual crowding was negligible in the Beery-VMI subtests. Furthermore, a similar visual–cognitive profile was recently found in studies on oculomotor and neurodevelopmental disorders without perceptual deficits, such as strabismus, oculomotor apraxia, and autism [71,73,74]. As suggested by Braddick and Atkinson, a transnosographic “dorsal stream vulnerability” could be responsible for alterations in visual–motor control and spatial cognition in many different developmental disorders [67]. Recently, Bathelt et al. reported that the development of visual brain areas, and specifically the dorsal stream, may be disrupted due to a perceptual alteration, thus finding an anatomical substrate for such a functional involvement and relating this to sensory deprivation [18]. Based on these hypotheses and on our findings, we could argue that the chronic instability of foveation in IIN may affect the functions of the dorsal stream. In conclusion, children with IIN seem to share both cognitive and visual–cognitive peculiarities with neurodevelopmental disorders. We are certainly far from understanding the causes of such a condition. Moreover, we cannot completely rule out that nystagmus and its peculiar neuropsychological profile are both symptoms of a common underlying condition that current tools cannot detect. Nevertheless, our results may have important implications for the management of such condition. For example, by considering the relation between VMI and learning abilities [68,69], a screening for learning disabilities should be proposed for children with IIN. A total of 20 children from our cohort performed a specific assessment that revealed frailties in at least one competence (reading, writing, or math) in the majority. An early assessment, including visual functions (perceptual and oculomotor aspects) and visual–motor integration, is fundamental to detecting a subclinical vulnerability and providing dedicated training [49,75]. Daibert-Nido and colleagues, for example, proposed a specific type of oculomotor training in children with IIN using biofeedback technologies with a positive impact not only on visual functions, but also on reading speed and quality of life [76]. Further studies are needed to investigate possible intervention strategies in this population.

Finally, this study explored whether basic visual functions (BCVA for far and near distance, fixation, smooth pursuit, and saccades) would influence neuropsychological and academic aspects, and no significant associations were found. This may be at least partially explained by our relatively small sample. Future studies on larger cohorts should evaluate the relative weight of oculomotor and perceptual impairment on the observed neuropsychological frailties while considering bigger samples of patients, attending different school grades, and using homogeneous evaluation tools. Furthermore, our study is based on clinical evaluations that, being circumscribed in time and contemplating breaks, are the expression of a child’s best performance and do not consider, for instance, ocular fatigability, an aspect that could emerge in everyday life activities (e.g., when a child is required to spend long hours in school performing vision-related tasks) and affect attentional and cognitive performances. Thus, we believe more specific and ad hoc evaluations should be carried out in ecological settings, considering factors such as the general functional profile of a child, the fluctuations in his/her visual functioning, and the possible environmental adaptations. The results of such studies could shed light not only on the necessity and modality of assessment of the learning abilities in children with IIN (and on the possibility of a specific intervention), but also on the unclear functional mechanisms underlying this condition.

## 5. Conclusions

Deepening the study of IIN may both be helpful for a better and comprehensive assessment of such a disease in clinical settings and provide a window into the understanding of brain development.

First, even though global cognitive and visual–cognitive performances are in the normal range, children and adolescents with idiopathic infantile nystagmus express a peculiar cognitive and visual–cognitive profile, characterized by a relative deficit of processing speed, visual–motor integration, and motor tasks compared to purely perceptual, verbal, and reasoning tasks. This may reflect a subclinical vulnerability in such competencies that form the basis for effective learning and academic inclusion. Furthermore, long daily exposure to vision-related tasks may cause fatigue and an attentional decrease in children with IIN. Exploring the visual, cognitive, and visual–cognitive profiles of children with IIN, even in the absence of clinically evident deficits, may uncover specific struggles, allowing the design of tailored interventions and the implementation of environmental adaptations in school and social contexts.

Finally, children with IIN seem to share the same cognitive and visual–cognitive profiles with different neurodevelopmental disorders that may be due to a specific and still poorly known dorsal stream vulnerability. Given the known association between visual impairment and neurodevelopmental disorders, the neurodevelopmental trajectories of children with IIN deserve further investigation.

## Figures and Tables

**Figure 1 brainsci-13-01348-f001:**
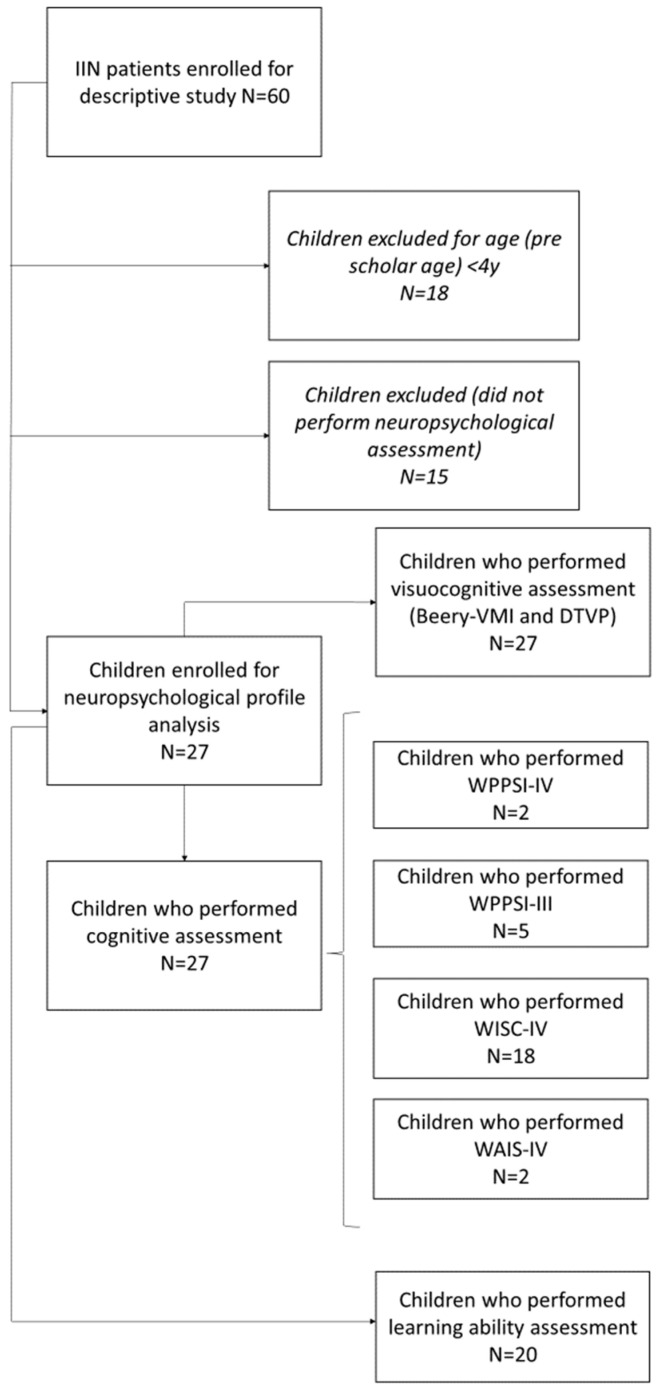
Flow chart presenting the selection process for neuropsychological assessment. WPPSI-III and IV: Wechsler preschool and primary scale of intelligence, third and fourth edition. WISC-IV: Wechsler intelligence scale for children, fourth edition. WAIS-IV: Wechsler adult intelligence scale, fourth edition. Beery-VMI: Beery developmental test of visual–motor integration. DTVP: developmental test for visual perception.

**Figure 2 brainsci-13-01348-f002:**
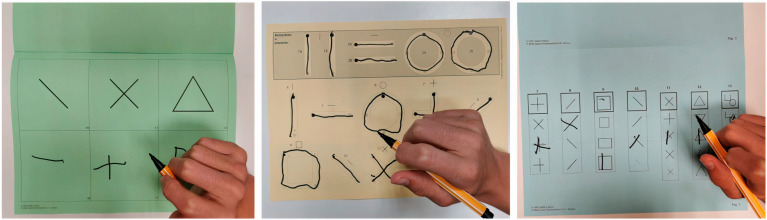
Beery-VMI tasks. Left to right: visual–motor integration (VMI), motor coordination (VMI-M), and visual perception (VMI-V). The VMI task requires the imitation and copying of a developmentally sequential series of geometric forms. In VMI-M, the subject is asked to trace the stimulus shape with a pencil without leaving the edges of the printed path. In the VMI-V task, the subject is asked to recognize the stimulus between similar choices.

**Figure 3 brainsci-13-01348-f003:**
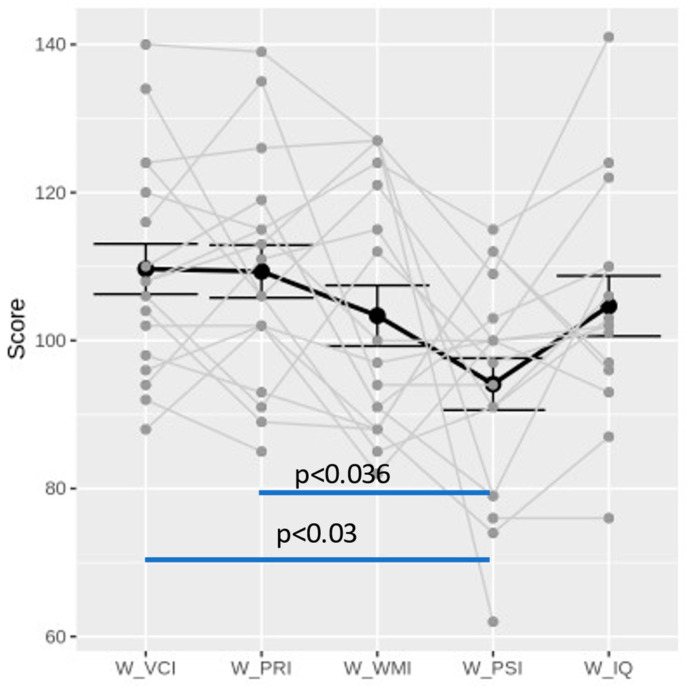
Graphic representation of IQ profile distribution of WISC-IV testing. W_VCI: verbal comprehension index. W_PRI: perceptual reasoning index. W_WMI: working memory index. W_PSI: processing speed index. W_IQ: intelligence quotient. Blue lines highlight the significance level of differences between indices.

**Figure 4 brainsci-13-01348-f004:**
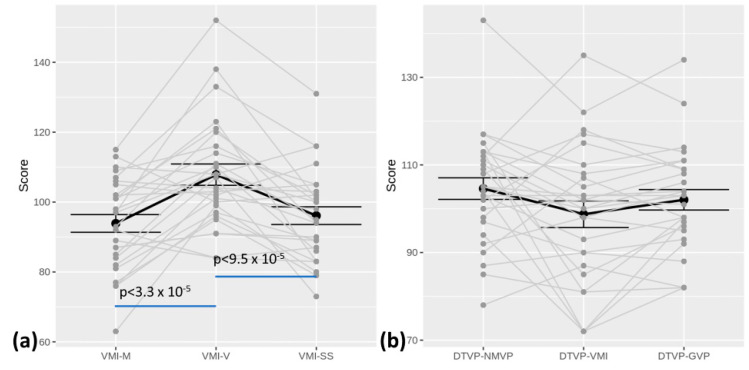
(**a**) Beery-VMI profile distribution; (**b**) DTVP profile distribution. Blue lines highlight the significance level of differences between indices.

**Table 1 brainsci-13-01348-t001:** Demographic characteristics of the cohort.

Parameter		*n*	%
Sex	F	17	28.3
	M	43	71.7
Age	Mean: 7.0 Y	Median: 6.9 Y	Range: 0.2–17.9 Y

**Table 3 brainsci-13-01348-t003:** WISC-IV testing results. A total of 18 patients underwent a WISC-IV cognitive assessment. W_VCI: verbal comprehension index. W_PRI: perceptual reasoning index. W_WMI: working memory index. W_PSI: processing speed index. W_IQ: intelligence quotient.

WISC-IV (N = 18)	**Index**	**N Performed/Interpretable**	**Mean**	**Median**	**Range**	**Standard Deviation**
	W_VCI	18	109.7	108	88–140	14.4
	W_PRI	18	109.3	108.5	85–139	15.1
	W_WMI	17	103.4	97	82–127	16.9
	W_PSI	18	94.1	98.5	62–115	14.9
	W_IQ	15	104.7	102	76–141	15.8

**Table 4 brainsci-13-01348-t004:** Visual–cognitive testing results. A total of 27 patients (17 M, 10 F, mean age: 9.5 y, range: 4.3–17.8 y) underwent the visual–cognitive assessment as explained in the methods section, according to their age. Beery-VMI = visuomotor integration test, VMI-V = visual perception, VMI-M = motor coordination, VMI-SS = visuomotor integration, DTVP = developmental test for visual perception, GVP = general visual perceptual, NMVP = non-motor visual–perceptual, and VMI = visual–motor integration.

Visual–Cognitive Test	Subtest	Mean	Median	Range	SD	Category	Count (%)
Beery-VMI (*n* = 27)	VMI-V	107.9	104	84–152	13.2	deficit (<5°p)	0
						frailty (5°–16°p)	2
						normal (>16°p)	25
	VMI-M	93.9	97	63–115	15.9	deficit (<5°p)	1
						frailty (5°–16°p)	6
						normal (>16°p)	20
	VMI-SS	96.1	95	73–131	13.2	deficit (<5°p)	1
						frailty (5°–16°p)	3
						normal (>16°p)	23
DTVP (*n* = 27)	GVP	102.0	103	82–134	12.1	deficit (<5°p)	0
						frailty (5°–16°p)	3
						normal (>16°p)	24
	MRVP	104.6	105	78–143	15.8	deficit (<5°p)	0
						frailty (5°–16°p)	1
						normal (>16°p)	26
	VMI	98.7	100	35–72	12.1	deficit (<5°p)	3
						frailty (5°–16°p)	2
						normal (>16°p)	22

**Table 5 brainsci-13-01348-t005:** Learning ability assessment test results.

Learning Ability Test	Normal (%)	Deficit (%)
Reading speed (*n* = 20)	15 (75)	5 (25)
Reading accuracy (*n* = 20)	18 (90)	2 (10)
Text comprehension (*n* = 19)	15 (79)	4 (21)
Writing (*n* = 16)	14 (88)	2 (12)
Math (*n* = 20)	14 (70)	6 (30)
Total (*n* = 20)	7 (35)	13 (65)

## Data Availability

Raw data supporting the results of this work will be available at the link 10.5281/zenodo.8359158.

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
