# Peer review of "Visual Function and Neuropsychological Profiling of Idiopathic Infantile Nystagmus"

_brainsci, 2023, doi:10.3390/brainsci13091348_

Round 1

Reviewer 1 Report

Re: “Visual function and neuropsychological profiling of Idiopathic Intantile Nystagmus” by Morelli et al.

This is an ambitious study of the neuro-psychological profile in a group diagnosed with Idiopathic Infantile Nystagmus (IIN). Already here I have a problem as the authors seem to think that IIN is one diagnose with a not yet understood etiology. In my experience INN is a heterogenous group with various etiologies. The number of individuals that get diagnosed with IIN depends on the quality of work up. The use of more advanced imaging will reduce the number of children with IIN.

In the introduction I would like the authors to add information about sensory and neurologic causes of infantile nystagmus. One such cause is periventricular white matter damage of immaturity. And after such presentation describe criteria for IIN

Participants. 60 subjects with IIN were recruited. Which examinations had been performed to motivate the diagnose of IIN? Were all 60 carefully examined in the slit lamp excluding pathologies of the anterior segment? Were they imaged with retinal OCT ruling out those with hypoplasia of the macula, or hypoplasia of the disc, or those with thinning of the retinal ganglion cell layer due to trans-synaptic degeneration from a retro-geniculate lesion? Were they all examined with ERG to exclude those with degenerative retinal disease? Did they all have normal anterior and posterior visual pathways and normal visual cortex as seen with conventional MRI of the brain? What about genetic testing? How many were born preterm? How come that a majority of the group were boys?

Discussion: The authors could add to the discussion that there is a possibility that the neuro-psychological profile found in these children could be not only caused by nystagmus interfering with neurodevelopment but also that the neuro-psychological profile is associated to nystagmus, both clinical signs caused by the same cerebral pathology. Dorsal stream dysfunction and infantile nystagmus may be found in populations of children born preterm.

I appreciate the last sentences of the conclusions.

Author Response

Re: “Visual function and neuropsychological profiling of Idiopathic Intantile Nystagmus” by Morelli et al.

This is an ambitious study of the neuro-psychological profile in a group diagnosed with Idiopathic Infantile Nystagmus (IIN). Already here I have a problem as the authors seem to think that IIN is one diagnose with a not yet understood etiology. In my experience INN is a heterogenous group with various etiologies. The number of individuals that get diagnosed with IIN depends on the quality of work up. The use of more advanced imaging will reduce the number of children with IIN.

In the introduction I would like the authors to add information about sensory and neurologic causes of infantile nystagmus. One such cause is periventricular white matter damage of immaturity. And after such presentation describe criteria for IIN.

R: We’d like to thank you for your precious comment, that allowed us to better clarify the Infantile Nystagmus classification. We referred to articles such as Papageorgiu et al. and Casteels et al. for a better clarification of the definition of infantile nystagmus, specifying that Idiopathic Infantile Nystagmus (IIN) is a form of Infantile Nystagmus and distinguishing it from forms with a known etiology (i.e., Sensory deficit Nystagmus and Neurological Nystagmus). We explained in the text that our diagnosis is a diagnosis of exclusion until proven otherwise and that furthers examinations may reveal subclinical sensory or neurologic deficits (lines 37-54).

Participants. 60 subjects with IIN were recruited. Which examinations had been performed to motivate the diagnose of IIN? Were all 60 carefully examined in the slit lamp excluding pathologies of the anterior segment? Were they imaged with retinal OCT ruling out those with hypoplasia of the macula, or hypoplasia of the disc, or those with thinning of the retinal ganglion cell layer due to trans-synaptic degeneration from a retro-geniculate lesion? Were they all examined with ERG to exclude those with degenerative retinal disease? Did they all have normal anterior and posterior visual pathways and normal visual cortex as seen with conventional MRI of the brain? What about genetic testing? How many were born preterm? How come that a majority of the group were boys?

R: Thank you for your comment. We modified the Participants section adding lines 107-114.

Being IIN a diagnosis of exclusion, we agree on the importance of a specific focus on diagnostic examinations performed in order to rule out a sensory or neurologic cause of nystagmus. For better clarification, we added a Supplementary Table summarizing all diagnostic examinations our patients underwent during the follow up in our Centre (see Supplementary File 2). No patients showed anterior segment and fundus oculi alterations, neither signs of neurological/CNS disorders at the neuropediatric and ophthalmological visits during follow-up. ERG examination was available for 53/60 patients and resulted normal for 90% of them. 10% of patients did not cooperate during the test. Moreover, no clinical and ophthalmological signs of retinal deterioration were reported at the follow-up both of the tested and untested patients for electroretinogram. A VEP examination was performed in 53/60 patients. 15/53 resulted completely normal, while 37/51 resulted mildly prolonged. Aspecific and inconstant VEP alteration has been previously reported by some authors in children with IIN (see for example Saunders et al. or Brecelj et al).

Brain MRI is not a routinary examination in the absence of clinical signs of degenerative or neurologic disease. Moreover, considering that for infants and children a narcosis is often  required, most of our cohort did not perform a brain MRI (consider that the mean age of the sample is 7.0yo). However, 24 patients in our cohort performed brain MRI on a 3T scanner, including DTI sequences, and none of them revealed significant alterations. Our cohort included 7 patients who were born slightly premature. 3 of them performed a brain MRI, which did not show any sign of CNS involvement.

OCT is not a routinary available examination for IIN patients, unless they show signs of degenerative macular/retinal disease. Moreover, in infancy, the OCT technique is also not easy to perform and less reliable [Suppiej et al]. Indeed, previous works diagnosed patients with IIN even without OCT confirmation [see for example Brecelj et al, 2004 doi:10.1016/j.clinph.2003.10.011. Thus, despite we agree considering that OCT would play a pivotal role to exclude any macular/retinal disease with greater confidence, the clinical and electrophysiological findings and based on current literature, of all our patients led us to consider them presumably affected from IIN even in the absence of an OCT screening, until proven otherwise.

23 patients performed genetic testing, 6 of which resulted with a pathogenic mutation in FRMD7. All genetic testing consisted of NGS testing (WES and/or gene panels) including a screening for the known causative genes of IIN and, in case of negative results, widened to the known genes responsible of retinal dystrophies, macular/foveal hypoplasia.

Your question about the prevalence of male patients in our cohort is interesting and challenging. Actually, we do not know the reason of such prevalence. However, we may hypothesize a possible correlation with an X-linked transmission of IIN, even in the already screened- and negative-patients. This consideration remains a pure speculation, and we would be grateful if you may suggest us possible interpretations about such datum. 

Discussion: The authors could add to the discussion that there is a possibility that the neuro-psychological profile found in these children could be not only caused by nystagmus interfering with neurodevelopment but also that the neuro-psychological profile is associated to nystagmus, both clinical signs caused by the same cerebral pathology. Dorsal stream dysfunction and infantile nystagmus may be found in populations of children born preterm.

R: Thank you for your precious observation, we integrated it in the discussion (lines 363-365)

I appreciate the last sentences of the conclusions.

R: Thank you!

Reviewer 2 Report

This is an interesting paper which aims to carry out neuropsychological profiling in children with idiopathic infantile nystagmus. There is a relative paucity of work in this area - so this would be of interest to varied audience of researchers, patient groups, educators etc. However, there are several areas that require clarification/improvement in this manuscript.

In the first paragraph of the introduction, the definitions of infantile nystagmus syndrome, idiopathic and congenital nystagmus appear to be confused when describing the epidemiology. The paper that has been quoted reports the prevalence as: " For the total population (children and adults combined), the prevalence of unassociated INS was 1.9 per 10,000 population (95% CI, ±1.6), INS associated with albinism was 2.5 per 10,000 population (95% CI, ±0.9), INS associated with retinal diseases was 3.4 per 10,000 population (95% CI, ±2.1), INS associated with low vision was 4.2 per 10,000 population (95% CI, ±1.2), and FMNS was 0.6 per 10,000 population (95% CI, ±0.4). The total prevalence for INS was 14.0 per 10,000 population (95% CI, ±3.1; 12.0 ± 0.9 per 10,000 18 years of age or younger and 14.7 ± 3.8 per 10,000 older than 18)." I cannot locate the figure of 60% of cases being idiopathic in newborn - which seems rather high. Please clarify.

This is described as a retrospective study. Does this centre routinely carry out neuropsychological assessments on all children presenting with nystagmus as part of standard clinical care? Some of these tests can take quite a bit of time and some of the cohort are very young (2 years is the youngest) and I wonder if it was possible to do these in conjunction with all of the standard of care ophthalmological assessments as well as part of their clinical visits. Or were these tests carried out later on on a separate day/time or as part of a clinical/research study - in which case ethical/institutional approval would normally be required? 

Clarification with regards to the exclusion criteria is required. The reference provided makes it clear that idiopathic infantile nystagmus is a diagnosis of exclusion. Therefore, full phenotyping would include in addition to clinical examination, optical coherence tomography examinations to exclude retinal and optic nerve morphological abnormalities, ISEV standard VEPs and ERGs to exclude optic nerve misrouting and retinal dystrophies and genetic testing. Did all cases undergo full phenotyping before a diagnosis of IIN was made? The reason I ask is that a rather high number of subjects 45% are reported to have severe low vision in this study - which is not typical in IIN - where visual acuity is usually well preserved.

With regards to the neuro-psychological tests that were carried out, it seems that only certain subsets of subjects were able to carry these out. It is not clear which subjects carried out each tests or if some subjects carried out all of the tests. These makes interpreting the results/discussion of this report difficult as a reader. In some cases e.g. very young children I'm certain that none of the tests would have been possible to carry out with any degree of reliability. In which case minimum age should be included as an exclusion factor. In order to get the full benefit of interpreting these tests, the visual function and phenotypic characteristics of the specific group(s) that participated in these tests need to be clear. Were there any significant differences in the patients that performed the tests and the patients who didn't performed the tests that could have confounded the results/conclusions of the study?

I noted a typo table 2 - where pendular was spelled as pendolar. There may be other small errors elsewhere (please check for these) but otherwise, good standard of English language writing.

Author Response

This is an interesting paper which aims to carry out neuropsychological profiling in children with idiopathic infantile nystagmus. There is a relative paucity of work in this area - so this would be of interest to varied audience of researchers, patient groups, educators etc. However, there are several areas that require clarification/improvement in this manuscript.

In the first paragraph of the introduction, the definitions of infantile nystagmus syndrome, idiopathic and congenital nystagmus appear to be confused when describing the epidemiology.

The paper that has been quoted reports the prevalence as: " For the total population (children and adults combined), the prevalence of unassociated INS was 1.9 per 10,000 population (95% CI, ±1.6), INS associated with albinism was 2.5 per 10,000 population (95% CI, ±0.9), INS associated with retinal diseases was 3.4 per 10,000 population (95% CI, ±2.1), INS associated with low vision was 4.2 per 10,000 population (95% CI, ±1.2), and FMNS was 0.6 per 10,000 population (95% CI, ±0.4). The total prevalence for INS was 14.0 per 10,000 population (95% CI, ±3.1; 12.0 ± 0.9 per 10,000 18 years of age or younger and 14.7 ± 3.8 per 10,000 older than 18)." I cannot locate the figure of 60% of cases being idiopathic in newborn - which seems rather high. Please clarify.

R: Thank you for you precious comment. We noticed an error in data interpretation of such article and proceeded with its correction. Moreover, we widened the section containing epidemiologic data of IIN, reporting also more recent data on both prevalence and incidence. [Lines 37-54]

This is described as a retrospective study. Does this centre routinely carry out neuropsychological assessments on all children presenting with nystagmus as part of standard clinical care? Some of these tests can take quite a bit of time and some of the cohort are very young (2 years is the youngest) and I wonder if it was possible to do these in conjunction with all of the standard of care ophthalmological assessments as well as part of their clinical visits. Or were these tests carried out later on on a separate day/time or as part of a clinical/research study - in which case ethical/institutional approval would normally be required? 

R: Thank you for your comment that allows us to present the standard evaluation protocol of our Centre. To rule out a negative impact of visual impairment on different areas of development and to sustain academic performances, we routinely carry out a thorough neuropsychological assessment. Such a protocol has a primary clinical purpose and is adapted to the visual, cognitive, and global profile of each child, as well as on his/her age. Thus, the data we present derive from children who were able (according to their age, scholarization, visual acuity, global cooperation level, and so on) to perform the mentioned tests. At their first admission, the parent/legal guardian of each child signs an informed consent allowing the researchers to use data from clinical evaluation for research after their anonymization.

To clear the inclusion of neuropsychological data for this paper, we designed a flow chart that you can find in the manuscript (Figure 1).

Clarification with regards to the exclusion criteria is required. The reference provided makes it clear that idiopathic infantile nystagmus is a diagnosis of exclusion. Therefore, full phenotyping would include in addition to clinical examination, optical coherence tomography examinations to exclude retinal and optic nerve morphological abnormalities, ISEV standard VEPs and ERGs to exclude optic nerve misrouting and retinal dystrophies and genetic testing. Did all cases undergo full phenotyping before a diagnosis of IIN was made? The reason I ask is that a rather high number of subjects 45% are reported to have severe low vision in this study - which is not typical in IIN - where visual acuity is usually well preserved.

R: Thanks for your comment. We agree that a better clarification of inclusion criteria was necessary, so we added diagnostic work-up description in “Participants” section (lines 107-114). Moreover, a more accurate description of our cohort is now available as supplementary file 2. Considering visual acuity in our cohort, we were quite surprised by these data as well as we agree it is not typical. We hypothesize they may be partly due to crowding effect and the immaturity of compensation strategies of younger children (e.g., head turn), but any suggestion for other possible interpretations would be precious. Anyway, all children with worse visual acuity performed an ERG (in addition to complete ophthalmological evaluations) that ruled out retinal involvement.

With regards to the neuro-psychological tests that were carried out, it seems that only certain subsets of subjects were able to carry these out. It is not clear which subjects carried out each tests or if some subjects carried out all of the tests. These makes interpreting the results/discussion of this report difficult as a reader. In some cases e.g. very young children I'm certain that none of the tests would have been possible to carry out with any degree of reliability. In which case minimum age should be included as an exclusion factor. In order to get the full benefit of interpreting these tests, the visual function and phenotypic characteristics of the specific group(s) that participated in these tests need to be clear. Were there any significant differences in the patients that performed the tests and the patients who didn't performed the tests that could have confounded the results/conclusions of the study?

R: Thank you for your comment. Being a retrospective study, we included the neuropsychological profiles of all patients who underwent the cognitive and visuo-cognitive assessment. A group of patients resulted not eligible a priori because of their age (<4 years). Remaining patients did not show any significant clinical or neurophysiological differences, and their enrolment only relied if the assessment was performed or less. For clarification, anyway, see the flow chart in the main text (Figure 1).

I noted a typo table 2 - where pendular was spelled as pendolar. There may be other small errors elsewhere (please check for these) but otherwise, good standard of English language writing.

R: Thank you, we corrected the mentioned typo and checked for others throughout the manuscript.

Round 2

Reviewer 1 Report

I am satisfied with answers and changes. Children with a developmental age of 4 years often cooperate well in OCT. For your next study....